# Rational Photodeposition of Cobalt Phosphate on Flower-like ZnIn_2_S_4_ for Efficient Photocatalytic Hydrogen Evolution

**DOI:** 10.3390/molecules29020465

**Published:** 2024-01-17

**Authors:** Yonghui Wu, Zhipeng Wang, Yuqing Yan, Yu Wei, Jun Wang, Yunsheng Shen, Kai Yang, Bo Weng, Kangqiang Lu

**Affiliations:** 1Jiangxi Provincial Key Laboratory of Functional Molecular Materials Chemistry, School of Chemistry and Chemical Engineering, Jiangxi University of Science and Technology, Ganzhou 341000, China; 2cMACS, Department of Microbial and Molecular Systems, KU Leuven, 3001 Leuven, Belgium

**Keywords:** photocatalytic H_2_ evolution, indium zinc sulfide, cocatalyst, cobalt phosphate, photogenerated holes transfer

## Abstract

The high electrons and holes recombination rate of ZnIn_2_S_4_ significantly limits its photocatalytic performance. Herein, a simple in situ photodeposition strategy is adopted to introduce the cocatalyst cobalt phosphate (Co-Pi) on ZnIn_2_S_4_, aiming at facilitating the separation of electron–hole by promoting the transfer of photogenerated holes of ZnIn_2_S_4_. The study reveals that the composite catalyst has superior photocatalytic performance than blank ZnIn_2_S_4_. In particular, ZnIn_2_S_4_ loaded with 5% Co-Pi (ZnIn_2_S_4_/5%Co-Pi) has the best photocatalytic activity, and the H_2_ production rate reaches 3593 μmol·g^−1^·h^−1^, approximately double that of ZnIn_2_S_4_ alone. Subsequent characterization data demonstrate that the introduction of the cocatalyst Co-Pi facilitates the transfer of ZnIn_2_S_4_ holes, thus improving the efficiency of photogenerated carrier separation. This investigation focuses on the rational utilization of high-content and rich cocatalysts on earth to design low-cost and efficient composite catalysts to achieve sustainable photocatalytic hydrogen evolution.

## 1. Introduction

Rapid economic and social development depends on fossil fuels. However, due to the non-renewable nature of fossil fuels and the detrimental impact on the environment, it is imperative that we urgently seek sustainable energy sources capable of replacing them [1,2,3,4,5]. Hydrogen (H_2_) energy, as a clean and renewable energy source, is one of the most promising alternative energy sources for fossil fuels [6,7,8]. Among various H_2_ production methods, solar-driven water splitting for H_2_ production is considered as a green and sustainable solar energy conversion technology, which can relieve the pressure of energy dilemma and environmental pollution [9,10,11,12]. Consequently, there is an urgent need to develop photocatalysts with high performance to promote the application of photocatalytic H_2_ evolution technology [13]. Nowadays, due to their remarkable light absorption properties and special electronic structures, metal sulfides have become a hot topic in the field of solar energy conversion technology.

As a ternary sulfide, ZnIn_2_S_4_ has attracted global attention from researchers on account of its favorable layered structure, simple synthesis, good photostability and suitable electronic band structure [14,15]. In particular, the flower-like structure has a high surface area and improves the light absorption through multiple reflections, which plays an important role in enhancing the photocatalytic performance [16,17,18]. However, due to the high recombination rate of photogenerated electron–hole pairs, pure ZnIn_2_S_4_ exhibits low photocatalytic activity [19,20,21,22]. To address this problem, the rational introduction of cocatalyst is a viable approach to optimize the activity and stability of ZnIn_2_S_4_ [23]. Among the many cocatalysts, cobalt phosphate (Co-Pi) has demonstrated remarkable ability to transfer photogenerated holes from different light-collecting semiconductors in previous studies and has been reported to improve their overall performance [24]. Therefore, the rational introduction of the holes cocatalyst Co-Pi into ZnIn_2_S_4_ is expected to obtain a cost-effective and efficient composite photocatalyst to promote photocatalytic H_2_ evolution. Moreover, in situ photodeposition is considered to be a promising method to enhance the photocatalytic activity of semiconductors, due to its advantages such as close contact, simple preparation and directional loading [25,26,27]. Consequently, rationally introducing Co-Pi into ZnIn_2_S_4_ by in situ photodeposition is expected to promote the migration of photogenerated holes of ZnIn_2_S_4_, thereby improving the photocatalytic performance of the composite photocatalyst.

Herein, we prepare the ZnIn_2_S_4_ nanoflower substrate material by the hydrothermal method, and the hybrid catalyst is constructed by in situ photodeposition of cobalt phosphate (Co-Pi) on ZnIn_2_S_4_ nanoflower. The ZnIn_2_S_4_/Co-Pi composite exhibits a significantly enhanced performance in the photocatalytic H_2_ evolution compared to pure ZnIn_2_S_4_. Notably, the optimal ZnIn_2_S_4_/5%Co-Pi photocatalytic H_2_ production rate is 3593 μmol·g^−1^·h^−1^, which surpasses most similar hybrid cocatalyst systems reported in the literature (Table 1). The photo/electrochemical tests and photoluminescence (PL) confirm that the photogenerated carrier separation efficiency of the composite catalyst is significantly improved. This work aims to provide insights for designing cost-effective and efficient mixed catalysts to enhance overall photocatalytic performance through rationally exploiting earth-abundant cocatalysts.

## 2. Results and Discussion

The preparation process diagram of the ZnIn_2_S_4_/Co-Pi (ZIS/Co-Pi) composite is shown in Figure 1a. Initially, ZnIn_2_S_4_ (ZIS) nanoflower is prepared by a one-step hydrothermal process. Subsequently, Co-Pi is introduced to ZIS nanoflower by in situ photodeposition to obtain ZIS/Co-Pi composites. Due to the best photocatalytic H_2_ production performance of ZnIn_2_S_4_/5%Co-Pi (Z5CP), we mainly discuss this proportion of the composites in the subsequent characterization. According to Appendix A, the color of ZIS nanoflower changes significantly before and after in situ photodeposition, with pure ZIS appearing as bright yellow, and Z5CP appearing as yellowish green. The morphology and microstructure of different samples are obtained by field emission scanning electron microscopy (FESEM). As depicted in Figure 1b, pure ZIS presents a spherical flower-like structure with a diameter of about 1 μm. The SEM image of Z5CP (Figure 1c) shows that Z5CP inherits the flower-like structure of ZIS. Notably, the flower-like structure can provide a number of active sites, and multiple layers of petals enable light to be reflected multiple times, which leads to enhanced light absorption [36,37]. In addition, the SEM image of Z5CP shows that the Co-Pi nanoparticles are highly dispersed, and no large Co-Pi particles were observed. As presented in Figure 1d, transmission electron microscopy (TEM) characterization further confirms the spherical flower-like structure of ZIS. Moreover, Figure 1e shows that the Co-Pi nanoparticles are attached to the ZIS nanoflower, proving the successful synthesis of Z5CP composites. As depicted in Figure 1f, the lattice distance of Z5CP is about 0.297 nm corresponding to the (104) crystal face of ZIS, and the Co-Pi synthesized by in situ photodeposition is amorphous. Furthermore, the EDS spectra (Appendix A) and the element mapping results (Figure 1g) confirm the existence of Zn, In, S, P, O, and Co elements in Z5CP. The spatial distribution of Zn, In, S, O, P, and Co elements in the elemental mapping images of Z5CP composite shows that Co-Pi grows uniformly on the surface of ZIS nanoflower.

The phase structure and crystallinity are analyzed by the X-ray diffraction (XRD) map. Figure 2a displays the XRD spectra of both ZIS and Z5CP. For ZIS, the strong diffraction peaks at 27.5° and 47.2° belong to the (102) and (110) faces of hexagonal ZIS (JCPDS No.65-2023) [38]. For Z5CP composites, the XRD diffraction curve closely resembles that of ZIS except that there is a faint peak at 55.6° belonging to the (202) face of hexagonal ZIS, indicating that ZIS remains a stable crystal structure after coupling with Co-Pi [39]. However, in the Z5CP composite, the characteristic diffraction peak of Co-Pi is not observed due to the amorphous nature of in situ photodeposition of Co-Pi [40,41]. The optical characteristics of the photocatalysts are analyzed by UV-visible diffuse reflection spectroscopy (DRS). As depicted in Figure 2b, the pure ZIS displays a clear absorption edge around 520 nm, indicating a band gap of about 2.44 eV [42]. Compared with pure ZIS, the absorption intensity of Z5CP hybrid in the visible range (520~750 nm) increases with the strong absorption of Co-Pi, indicating that the introduction of Co-Pi can improve the visible light response of ZIS. Moreover, Figure 2b shows that there is no significant shift in absorption edge for the Z5CP composite, indicating that the Co-Pi cocatalyst only deposits on the ZIS surface and does not bind with the crystal lattice.

The chemical composition and elemental states of Z5CP composite are further determined by X-ray photoelectron spectroscopy (XPS). As presented in Figure 3a, Zn, In, S, Co, and P elements exist in the hybrid products, which further demonstrates the successful photodeposition of Co-Pi on the surface of ZIS nanoflower. As shown in Figure 3b, the XPS spectrum of Zn 2p exhibits two distinct peaks at 1045 and 1022 eV, which correspond to the binding energies of Zn 2p_1/2_ and Zn 2p_3/2_ of Zn^2+^, respectively. From the XPS spectrum of In 3d (Figure 3c), two peaks that center on binding energies 452.4 and 444.8 eV are respectively associated with In 3d_3/2_ and In 3d_5/2_, which indicate the +3 state of In. Moreover, as presented in Figure 3d, the peaks of 162.9 and 161.7 eV belong to S 2p_1/2_ and S 2p_3/2_, confirming the presence of S^2−^. In the XPS spectrum of Co 2p (Figure 3e), the peak of Co 2p_3/2_ is at 781.3 eV (satellite peak at 784.3 eV), indicating the presence of Co^2+^ in the Z5CP composite [43,44,45]. In addition, the peak of P 2p (Figure 3f) at 133.5 eV indicates that P presents in the form of phosphate groups, which further proves the successful synthesis of Z5CP [46].

Photocatalytic H_2_ production is performed with triethanolamine (TEOA) as the hole scavenger, and the photocatalytic properties of pure ZIS and different proportions of ZIS/Co-Pi composites under visible light are investigated. Figure 4a is a diagram of the photocatalytic activity of ZIS and composite with 1%, 5%, and 10% Co-Pi (hereinafter shown as Z1CP, Z5CP, and Z10CP, respectively). As shown in Figure 4a, due to the fast photogenerated electron–hole recombination rate, the pure ZIS is less active and the H_2_ evolution rate is only 1832 μmol∙g^−1^∙h^−1^. After the introduction of Co-Pi cocatalyst, Z1CP, Z5CP, and Z10CP all show better H_2_ evolution performance compared with blank ZIS. With the increase in Co-Pi content, the hydrogen yield increases gradually. However, when the Co-Pi content increases further, the H_2_ evolution activity decreases, which may be due to the remarkable shielding effect of Co-Pi, thereby decreasing the photocatalytic active sites [47]. In particular, the Z5CP composite shows the highest H_2_ evolution rate (3593 μmol∙g^−1^∙h^−1^), approximately two times higher than that of ZIS alone. This can be attributed to the fact that in situ photodeposition of Co-Pi promotes the transfer of photogenerated holes and reduces the recombination rate of photogenerated carriers. As shown in Table 1, the Z5CP composite prepared in this work has optimal photocatalytic H_2_ production properties compared with the photocatalytic H_2_ production activities of some representative ZIS-based composites reported in recent years. In addition, the stability of Z5CP is tested by the cyclic test. As depicted in Figure 4b, after five cycles, no apparent deactivation has been observed for Z5CP composite, indicating the excellent stability of Z5CP composite.

Photo/electrochemical tests are used to further characterize material reducing capacity and photogenerated carrier transfer efficiency. Linear sweep voltammetry (LSV) is first used to determine the H_2_ evolution performance of ZIS and Z5CP samples. Figure 5a shows the polarization curve of ZIS and Z5CP composites. It can be seen that the overpotential of Z5CP is less than ZIS at the same current density, indicating that the H_2_ evolution performance of Z5CP is better than that of ZIS [48]. The kinetics of photocatalysis in different samples can be compared by the Tafel slope. As shown in Appendix A, the Tafel slope of the Z5CP composite (0.21 V/decade) is smaller than that of ZIS (0.24 V/decade), indicating the better reduction effect and interfacial charge transfer efficiency of Z5CP, which is consistent with the photocatalytic H_2_ production activity as well as other characterization results [49]. These results further demonstrate that Z5CP has faster reaction kinetics and excellent interface carrier separation efficiency. To study the charge separation and transfer of these ZIS/Co-Pi composites, instantaneous photocurrent (IT), electrochemical impedance spectroscopy (EIS) and steady-state photoluminescence (PL) spectra are measured on the ZIS and Z5CP samples [50]. As illustrated in Figure 5b, the optical current density of ZIS is small, indicating that the photogenerated carrier separation efficiency of ZIS is poor. However, it is found that after the introduction of Co-Pi, the optical current density of Z5CP is significantly improved compared with that of pure ZIS, indicating that Z5CP has better separation efficiency of electron (e^−^) and hole (h^+^) [51,52,53,54,55]. As shown in Figure 5c, the radius of curvature of Z5CP composite is smaller than ZIS, indicating that the charge transfer resistance of Z5CP is lower, which improves the separation and transfer rate of photogenerated carriers, thus enhancing the photocatalytic activity [56,57,58,59,60]. Furthermore, Figure 5d describes the steady−state photoluminescence (PL) spectra test of the sample. As shown in Figure 5d, the PL intensity of Z5CP is significantly lower than that of blank ZIS, indicating that the addition of cocatalyst Co-Pi effectively inhibits the recombination of photogenerated carriers [61,62,63,64,65]. Taken together, the results of these photo/electrochemical tests validate the improved separation and transfer of photogenerated charges in Z5CP, leading to the enhanced performance of photocatalytic H_2_ evolution.

The information of chemical reaction area of the blank ZIS and the composite material Z5CP is obtained by the cyclic voltammetry test (CV). Figure 6a,b show the cyclic voltammetry (CV) curves of the blank ZIS and Z5CP composites, respectively. As illustrated in Figure 6c, the double-layer capacitance of Z5CP composite (3.99 μF·cm^−2^) is significantly larger than ZIS (1.83 μF·cm^−2^), which strongly proves that Z5CP has more active sites area than ZIS [45]. In addition, the flat charged position (E_fb_) of the original ZIS is measured with Mott–Schottky (MS). Generally, the slope of the positive one indicates that the semiconductor is an intrinsic n-type semiconductor [51]. As can be seen from Figure 6d, ZIS belongs to the n-type semiconductor. Moreover, Appendix A shows the detailed fitting parameters of MS. According to the x-intercept of the block, its E_fb_ is determined to be −0.52 V (vs. Ag/AgCl). In general, the conduction band position of n-type semiconductors is about 0.2 V more negative than that of E_fb_ [66,67,68]. Therefore, the conduction charge position (E_CB_) of the ZIS is −0.72 V (vs. Ag/AgCl). From the formula E_NHE_ = E_Ag/AgCl_ + 0.20 V, the E_CB_ of ZIS is −0.52 V (vs. NHE). According to the band gap of ZIS (2.44 eV), the valence band potential (E_VB_) of ZIS is 1.92 V (vs. NHE).

Combined with the above experiments and characterization, we propose a viable mechanism for photocatalytic H_2_ production of Z5CP under visible light. As shown in Figure 7, under visible light irradiation, Z5CP effectively absorbs the photon energy, and then the electrons on the valence band (VB) are excited and transition to the conduction band (CB), and the corresponding positive electric holes are generated on the valence band (VB). The electron (e^−^) migrated to the semiconductor surface binds to the H^+^ adsorbed in water to form H_2_. However, ZIS has a high electrons and holes recombination rate; therefore, its photocatalytic activity is limited. Notably, Co-Pi has the excellent property of transferring photogenerated holes, and the holes of ZIS are transferred to Co-Pi and drive cycles to catalyze the Co^2+/3+^ → Co^4+^→ Co^2+/3+^ reaction [24]. At the same time, ZIS rapidly exports holes to oxidize the sacrificial reagent of triethanolamine (TEOA); therefore, the resulting photogenerated hole (h^+^) is effectively separated and consumed by it. Therefore, the photogenerated carrier separation efficiency of the composite photocatalyst Z5CP is improved, which allows more electrons to transfer to the catalyst surface to react with H^+^ to produce more H_2_. This is also the main factor for the significant improvement of the photocatalytic H_2_ evolution performance of Z5CP composite.

## 3. Experimental Section

### 3.1. Materials

Concentrated sulfuric acid (H_2_SO_4_), triethanolamine (C_6_H_15_NO_3_, TEOA), anhydrous ethanol (C_2_H_5_OH), N,N-dimethylformamide (C_3_H_7_NO), disodium hydrogen phosphate dihydrate (Na_2_HPO_4_·2H_2_O), and sodium dihydrogen phosphate tetrahydrate (NaH_2_PO_4_·4H_2_O) are supplied by Xilong Scientific Co., Ltd. (Shantou, China). Cobalt nitrate hexahydrate (Co(NO_3_)_2_·6H_2_O), cetyltrimethylammonium bromide (C_19_H_42_BrN, CTAB), zinc nitrate hexahydrate (Zn(NO_3_)_2_·6H_2_O), indium chloride tetrahydrate (InCl_3_·4H_2_O), and Nafion solution (5 wt%) (C_9_HF_17_O_5_S) are supplied by Sinopharm Chemical Reagent Co., Ltd. (Shanghai, China).

### 3.2. Synthesis of ZnIn_2_S_4_ (ZIS)

Typically, Zn(NO_3_)_2_·6H_2_O (304.2 mg), InCl_3_·4H_2_O (624.4 mg), and cetyltrimethyl-ammonium bromide (CTAB) (230.6 mg) were added to a beaker containing 20 mL of deionized water and magnetically stirred for 30 min. Then, the thioacetamide (604.8 mg) was added to a beaker containing 10 mL deionized water and mixed to the above solution. Afterwards, the mixture was added to a Teflon liner and stirred for 30 min, and the liner was transferred to stainless steel autoclave heating in an oven at 433 K for 16 h. After cooling, the products were separated by filtration and washed several times with deionized water and ethanol. The resulting samples were dried under vacuum at 333 K for 12 h. Ultimately, a bright yellow solid was obtained.

### 3.3. Synthesis of ZnIn_2_S_4_/Co-Pi (ZIS/Co-Pi)

In a typical experiment, the prepared 200 mL (0.1 mol/L) NaH_2_PO_4_ and 200 mL (0.1 mol/L) Na_2_HPO_4_ solution were mixed and adjusted with pH to around 7. Subsequently, 80 mL of neutral buffer was measured, and the calculated amount of Co(NO_3_)_2_·6H_2_O was added to make it evenly dispersed by ultrasound. Furthermore, 40 mg of ZnIn_2_S_4_ was weighed and introduced into the aforementioned system which was then sealed using a sealing ring with several ventilation holes. Then, the system was subjected to Ar gas flow under magnetic stirring for 30 min followed by irradiation from a xenon lamp while maintaining stirring for an additional duration of 60 min after sealing. After the photodeposition, the samples were filtered with deionized water, and the samples were obtained after vacuum drying at 333 K for 12 h. The loading amount of Co-Pi in ZIS/xCo-Pi was altered by changing the amount of Co(NO_3_)_2_·6H_2_O. In the experimental design, the loading ratios of deposited Co-Pi in ZnIn_2_S_4_ are 1%, 5%, and 10%, respectively.

### 3.4. Activity Evaluation of Photocatalytic H_2_ Evolution

Photocatalytic H_2_ production was performed in a 50 mL airtight quartz reactor. In the entire quartz reactor, 5 mg of the catalyst was dispersed into a solution containing 5 mL of deionized water and 1 mL of triethanolamine (TEOA). Before the reaction, high purity Ar was injected into the quartz reactor for 30 min to exhaust the residual air in the reactor. A 300 W xenon lamp (λ > 420 nm) was selected as the light source, and after 2 h of illumination, 1 mL of gas was extracted into the gas chromatograph (thermal conductivity detector TCD, Agilent Technologies GC 7820A, Santa Clara, CA, USA) to detect the hydrogen yield obtained after the reaction. In order to evaluate the stability of ZIS/Co-Pi composite, the photocatalyst was separated and centrifuged. The recovered photocatalyst is then subjected to a subsequent cycle under the same conditions.

### 3.5. Characterization Methods

The morphological characteristics were tested through scanning electron microscopy (SEM, FESEM ZEISS sigma 500, Oberkochen, Batenwerburg, Germany) and transmission electron microscopy (TEM, Jeol JEM-2100F instrument, Jeol, Akishima, Tokyo). The determination of crystal structures was determined by X-ray diffraction (XRD) with Cu Kα (λ = 0.15406 nm, Bruker D8 Advance, Billerica, MA, USA). The surface composition of the samples was determined by X-ray photoelectron spectrometer (XPS, Thermo Fisher, K-Alpha, Waltham, MA, USA). The UV-visible diffuse reflectance spectrometer (DRS, Shimadzu UV-2600, Kyoto, Japan) was used to test the optical response of the catalyst. Photoluminescence (PL) spectra were obtained using a spectrofluorometer (FLS 980, Edinburgh Instruments Ltd., Edinburgh, UK) with an excitation wavelength of 500 nm. Furthermore, all the electrochemical measurements of the photocurrent, the electrochemical impedance spectra (EIS), the Mott–Schottky (MS), cyclic voltammetry (CV), and linear sweep voltammetry (LSV) curves were carried out in the three-electrode cell, in which Ag/AgCl was used as a reference electrode, a Pt wire was used as a counter electrode, and an indium in oxide (ITO) conductive glass was used with the samples as a working electrode in 0.1 M Na_2_SO_4_ electrolyte (pH = 7.56), all measurements were carried out on CH instruments CHI-660E electrochemical workstation (Shanghai Chenhua CHI-660E, Shanghai, China).

## 4. Conclusions

In summary, we synthesize spherical ZnIn_2_S_4_ nanoflower substrate material by the hydrothermal method, and reasonably construct a novel photocatalyst of indium zinc sulfide/cobalt phosphate (ZnIn_2_S_4_/Co-Pi) hybrid photocatalyst by the in situ photodeposition method. In the presence of cocatalyst cobalt phosphate (Co-Pi), the hybrid photocatalyst shows outstanding photocatalytic hydrogen evolution performance. Through changing the photodeposition amount of Co-Pi, it is observed that the highest H_2_ production rate of indium zinc sulfide (ZnIn_2_S_4_/5% Co-Pi) loaded with 5% cobalt phosphate (Co-Pi) is 3593 μmol·g^−1^·h^−1^, which is significantly higher than that of pure ZnIn_2_S_4_. The steady-state photoluminescence (PL) and electrochemical impedance spectroscopy (EIS) of the photocatalyst show that ZnIn_2_S_4_/Co-Pi composite has weaker PL intensity and lower charge transport resistance than blank ZnIn_2_S_4_, demonstrating that the hybrid photocatalyst has faster electron transfer and charge separation. Simultaneously, the larger double-layer capacitance and smaller overpotential of catalyst indicate that ZnIn_2_S_4_/Co-Pi composite has larger active area and better hydrogen evolution performance. This work makes reasonable use of the earth-abundant cocatalysts to design low-cost and efficient composite catalysts to promote the prospect of photocatalytic hydrogen evolution.

## Figures and Tables

**Figure 1 molecules-29-00465-f001:**
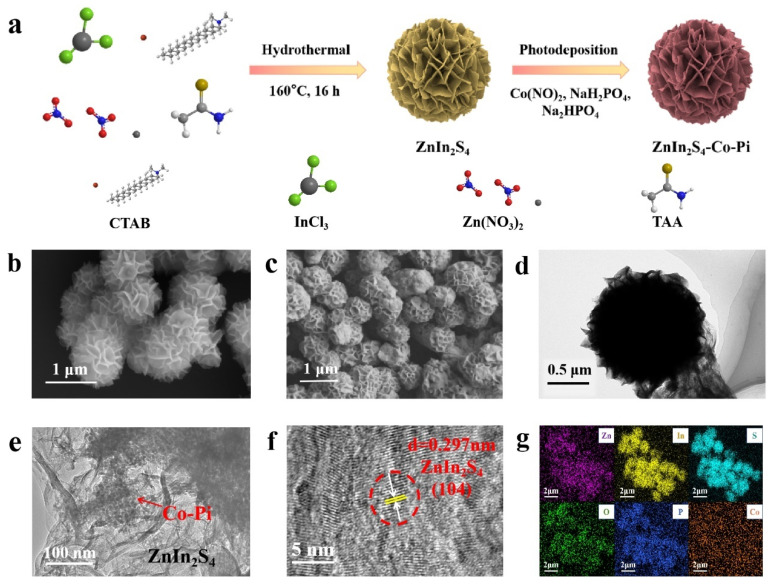
(**a**) Diagram illustrating the synthesis of ZIS/Co-Pi. (**b**,**c**) FESEM images of ZIS and Z5CP. (**d**–**f**) TEM images of Z5CP. (**g**) Mapping analysis results of Z5CP.

**Figure 2 molecules-29-00465-f002:**
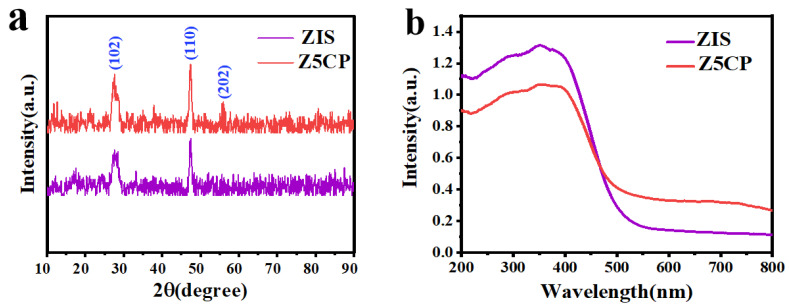
(**a**) X-ray diffraction (XRD) patterns and (**b**) UV–vis diffuse reflectance spectra (DRS) of ZIS and Z5CP.

**Figure 3 molecules-29-00465-f003:**
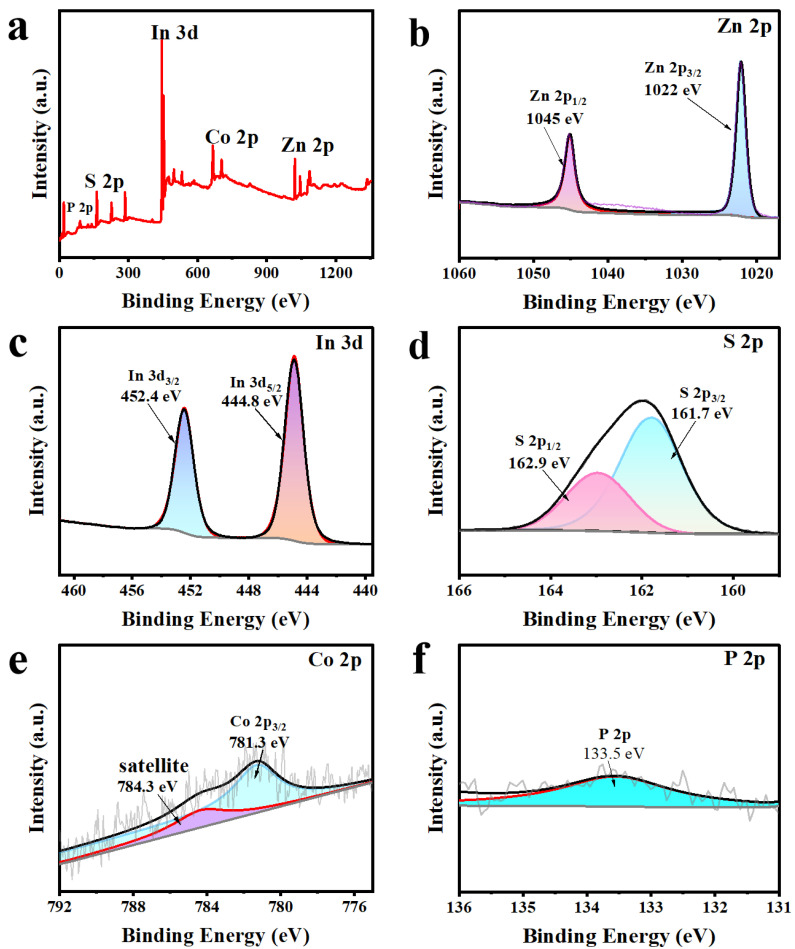
(**a**) XPS spectra of Z5CP, high-resolution spectra of (**b**) Zn 2p, (**c**) In 3d, (**d**) S 2p, (**e**) Co 2p, (**f**) P 2p.

**Figure 4 molecules-29-00465-f004:**
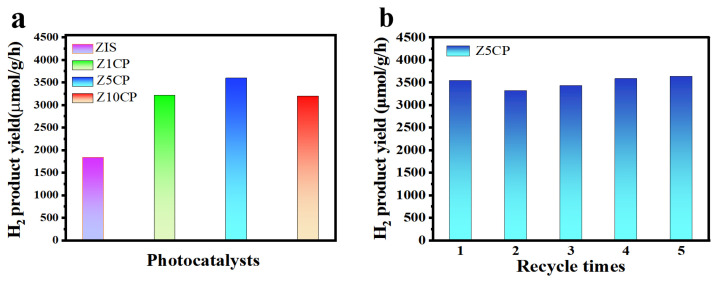
(**a**) Photocatalytic H_2_ production over pure ZIS and Z5CP composites. (**b**) Stability plots of the photocatalytic H_2_ production by Z5CP.

**Figure 5 molecules-29-00465-f005:**
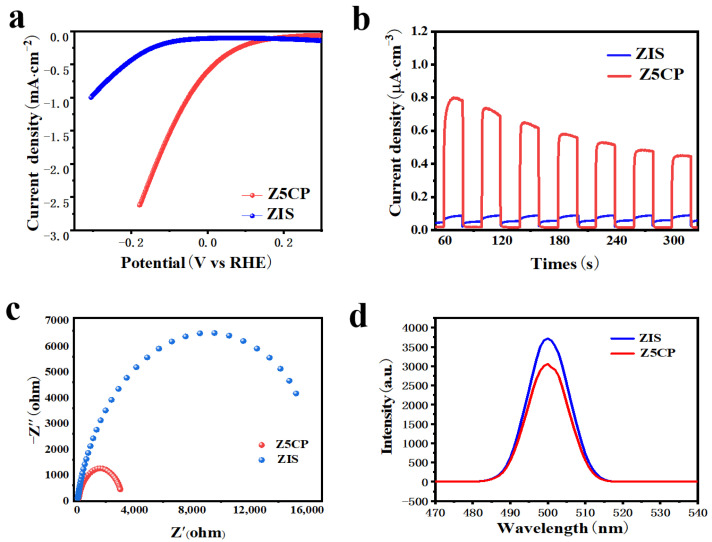
(**a**) Polarization curves. (**b**) Transient photocurrent spectra. (**c**) EIS Nyquist plots. (**d**) Steady−state photoluminescence (PL) emission spectra with an excitation wavelength of 500 nm.

**Figure 6 molecules-29-00465-f006:**
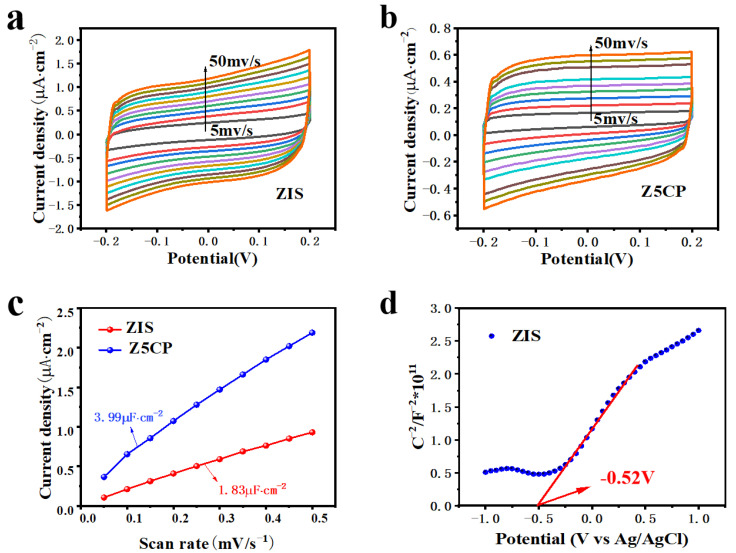
(**a**,**b**) Cyclic voltammetry curves of the ZIS and Z5CP. (**c**) Current density scan rate plot. (**d**) Mott−Schottky plots for ZIS.

**Figure 7 molecules-29-00465-f007:**
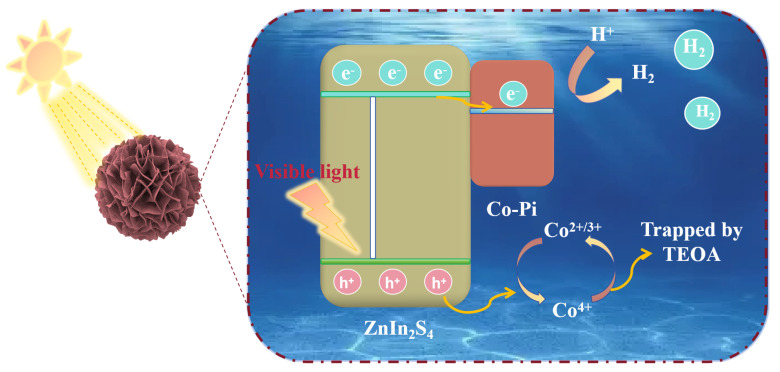
Mechanism diagram of Z5CP in the visible light-driven photocatalytic H_2_ production reaction.

**Table 1 molecules-29-00465-t001:** Comparison of the hydrogen production properties of the ZnIn_2_S_4_-based catalysts.

Photocatalysts	Light Sources	Sacrificial Agents	H_2_ (μmol∙g^−1^∙h^−1^)	Reference
ZnIn_2_S_4_-5%Co-Pi	300 W Xe lamp (λ ≥ 420 nm)	TEOA	3593	this work
ZnIn_2_S_4_/NiWO_4_	300 W Xe lamp (λ ≥ 420 nm)	TEOA	1781	[28]
ZnIn_2_S_4_/BPQDs	300 W Xe lamp (λ ≥ 420 nm)	TEOA	1207	[29]
J-ZnIn_2_S_4_/CdIn_2_S_4_	350 W Xe lamp (λ ≥ 420 nm)	TEOA	1830	[30]
N-ZnIn_2_S_4_	350 W Xe lamp (λ ≥ 400 nm)	Na_2_S/Na_2_SO_3_	262.62	[31]
MoO_2_/ZnIn_2_S_4_	300 W Xe lamp (λ ≥ 420 nm)	TEOA	2722.5	[32]
ReS_2_/ZnIn_2_S_4_	four 3 W 420 nm LED lamps	lactic acid (10 vol%)	2240	[33]
ZnIn_2_S_4_/CoFe_2_O_4_	300 W Xe lamp (λ ≥ 420 nm)	Na_2_S/Na_2_SO_3_	2260.5	[16]
NiCo_2_S_4_/ZnIn_2_S_4_	Xe lamp (λ > 400 nm)	-	770	[34]
CoS_1.097_/ZnIn_2_S_4_	300 W Xe lamp (780 nm ≥ λ ≥ 420 nm)	TEOA	2632.33	[35]

## Data Availability

Data are contained within the article.

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
