# Peer review of "Rational Photodeposition of Cobalt Phosphate on Flower-like ZnIn2S4 for Efficient Photocatalytic Hydrogen Evolution"

_molecules, 2024, doi:10.3390/molecules29020465_

Round 1
Reviewer 1 Report
Comments and Suggestions for Authors
This an interesting and well written article that is nearly ready for publication with very minor revisions. The following would improve the presentation:
Fig 3 a Tick marks are on the inside of this graph, where as all other graphs they are on the outside. To be consistent this should be changed.
Line 35-36 in the introduction states this material surpasses many literature values. Is a range of improvement possible? That way the reader can be aware of how much improvement has been achieved in this study.
Comments on the Quality of English LanguageIntroduction:
Line 13 "photostability stability" should be change to "photostability"
Results and Discussion
Line 84: "do not bind" should be changed to "does not bind"
Line 166 "about 0.2 V negative" should be changed to "about 0.2V more negative"
Experimental Section
Line 206 "Afterwards, after the mixture" should be changed to "Afterwards, the mixture"
Line 223 "changed by changing" may be better written as "altered by changing"
Reviewer 2 Report
Comments and Suggestions for Authors
In this study, the H2 production was investigated in the presence of ZnIn2S4 loaded with 5% Co-Pi (ZnIn2S4/5%Co-Pi) which demonstrated the best photocatalytic activity. The conclusion derived from the investigation is not sufficiently worthy that can attract broad readerships and can provide scientific insights. Therefore, the manuscript should be considerably revised to make more insights and show the scientific advancements. Following issues may be considered.
1. According to the authors «the optimal ZnIn2S4/5%Co-Pi photocatalytic H2 production rate is 3593 μmol·g-1·h-1, which surpasses most similar hybrid cocatalyst systems reported in the literature» (in Introduction part). It would be correct to cite literature data and operating conditions of other photocatalysts.
2. In this paper, the physical properties of the catalyst are mainly related to the catalytic performance of the catalyst, and some chemical properties of the catalyst are not explored in depth. In particular, the physicochemical properties of the best sample are discussed, but it is not clear why 5% gives such a result compared to 1% or 10%. In this case, the authors should probably start with the photocatalytic part and then describe the physicochemical properties of the best photocatalyst
3. XPS is a very powerful technique to quantify the Surface Composition which are essential to analyze the photocatalytic activity. Authors should do these detailed analysis.
Reviewer 3 Report
Comments and Suggestions for Authors
The authors of the paper " Rationally Photodeposition of Cobalt Phosphate on Flower-like ZnIn2S4 for Efficient Photocatalytic Hydrogen Evolution" have conducted valuable research on enhancing the electron-hole pair separation by facilitating the migration of photogenerated holes of ZnIn2S4 . The paper is well structured and coherent and could fit the molecule journal well. However, before making a final decision, I request the authors to address the following issues in a revised version.
1. The supplementary data are not attached to the paper, making authors are requested to include the supplementary data in the revised paper.
2. There are extra peaks in the Z5CP XRD data. However, the authors did not explain it in the paper. It is requested that the authors properly explain the additional peaks in the revised paper.
3. The XPS data shown in Fig. 3 (e) and (f) are very noisy. How confident are the authors on the fitting? The quality of the fitting parameters should be reported in the revised paper.
4. In Fig. 6 (d) the authors draw the line instead of linearly fitting the data, which can lead to incorrect conclusions. It is requested that the authors linearly fit the data and extract the parameters in the revised paper. It is also requested that the authors report the fitting parameters.
